# HIV risk behavior and associated factors among people living with HIV/AIDS in Ethiopia: A systematic review and meta-analysis

Yitayish Damtie[1], Bereket Kefale[1], Melaku Yalew[1], Mastewal Arefaynie[1],
Bezawit Adane[2], Amare Muche[2], Reta Dewau[2], Zinabu Fentaw[2], Erkihun
Tadesse Amsalu[2], Gedamnesh Bitew[3], Wolde Melese Ayele[2], Assefa Andargie Kassa[2],
Muluken Genetu Chanie[4], Mequannent Sharew Melaku[5], Metadel Adane[6] *

1 Department of Reproductive and Family Health, School of Public Health, College of Medicine and Health
Sciences, Wollo University, Dessie, Ethiopia, 2 Department of Biostatistics and Epidemiology, School of
Public Health, College of Medicine and Health Sciences, Wollo University, Dessie, Ethiopia, 3 Department of
Biostatistics and Epidemiology, Institute of Public Health, College of Medicine and Health Sciences,
University of Gondar, Gondar, Ethiopia, 4 Department of Health System and Policy, School of Public Health,
College of Medicine and Health Sciences, Wollo University, Dessie, Ethiopia, 5 Department of Health
Informatics, Institute of Public Health, College of Medicine and Health Sciences, University of Gondar,
Gondar, Ethiopia, 6 Department of Environmental Health, College of Medicine and Health Sciences, Wollo
University, Dessie, Ethiopia

☉ These authors contributed equally to this work.
* metadel.adane2@gmail.com

doi.org/10.1371/journal.pone.0269304

Deaconess Medical Center/Harvard Medical
School, UNITED STATES

**Data Availability Statement:** All relevant data are
within the manuscript and its Supporting
Information files.

## Abstract

### Background

HIV risk behavior among people living with HIV/AIDS (PLWHA) is a major public health concern as it increases HIV transmission. In Ethiopia, findings regarding HIV risk behavior have been inconsistent and inconclusive. Therefore, this meta-analysis aimed to estimate the pooled prevalence of HIV risk behavior and associated factors among PLWHA in Ethiopia.

### Methods

International databases, including Google Scholar, Cochrane library, HINARI, Pub Med, CINAHL, and Global Health were systematically searched to identify articles reporting the prevalence of HIV risk behavior and associated factors among PLWHA in Ethiopia. The data were analyzed using STATA/SE version-14. The random-effects model was used to estimate the pooled effects. I-squared statistics and Egger's test were used to assess the heterogeneity and publication bias respectively.

### Results

A total of 4,137 articles were reviewed and fourteen articles fulfilling the inclusion criteria were included in this meta-analysis. The pooled prevalence of HIV risk behavior in Ethiopia was 34.3%% (95% CI: 28.2, 40.3). Severe heterogeneity was observed between the included research articles ($I^2$ = 96.6, p = 0.000). Alcohol use (OR = 1.9, 95%, CI: [1.6, 2.3]),

**Funding:** The author(s) received no specific funding for this work.

**Competing interests:** The authors have declared that no competing interests exist.

**Abbreviations:** JBI, Joana Brigg Institute; MeSH, Medical Subject Headings; PLWHA, People Living with HIV/AIDS; PRISMA, Preferred Reporting Items for Systematic Reviews and Meta-Analysis; STI, Sexually Transmitted Infection; SSA, Sub-Saharan African.

HIV status non-disclosure (OR = 2.3, 95% CI: [1.3, 4.0]) and perceived stigma (OR = 2.3, 95% CI: [1.3, 4.1]) had a significant association with HIV risk behavior.

## Conclusion

The prevalence of HIV risk behavior among PLWHA in Ethiopia was high. Alcohol use, HIV status non-disclosure, and perceived stigma had a significant association with HIV risk behavior. In addition to promoting access to Antiretroviral Therapy (ART) treatment and improving medication adherence among PLWHA, various intervention programs focusing on the associated factors have to be implemented to tackle high-risk sexual behavior and go forward toward ending the HIV/AIDS pandemic.

## Background

HIV/AIDS remains a major public health challenge to the world population. In 2019, an estimated 37.9 million people were infected with HIV globally and 68% of patients were from sub-Saharan African (SSA) countries [1]. In Ethiopia, an estimated 722,248 people were living with HIV, and 14,872 people died from AIDS-related illnesses in 2017 [2].

The United States Centers for Disease Control and Prevention defines sexual risk behavior as a behavior that increases a person's risk of contracting sexually transmitted infections (STI) including HIV and/or experiencing unintended pregnancy [3]. It includes one or more of the following characteristics: sex without the use of condoms (condomless sex) [4–7], inconsistent condom use [5, 6, 8, 9], having multiple sexual partners [5, 6], sex with the influence of alcohol [5, 6], casual sex [5–7] and sexual exchange (exchange of money for sexual intercourse) [5, 6].

HIV risk behavior is frequently practiced by PLWHA. The result of the meta-analytic review showed that between 10% and 60% of PLWHA practiced condomless sex [10]. Reports suggested a high incidence of STI [11–13] and continued fertility desire among PLWHA on ART [14, 15], which indicate the practice of (or in the latter case, the need for) sexual intercourse unprotected by a condom. In SSA, more than 1 in 3 PLWHA practiced HIV risk behaviors [16–18]. In Ethiopia, the prevalence of sexual risk behaviors among PLWHA ranged from 15.8% to 56% [7, 19].

PLWHA following ART treatment show general improvements in their overall physical and clinical status and many of them believe that they are no longer infectious since they are on ART [20–22]. There is robust evidence that PLWHA with undetectable viral load cannot transmit HIV through sexual intercourse [23]. However, PLWHA may have challenges in accessing and adhering to ART, which can result in detectable viral loads and the potential for HIV transmission [24–27].

Various studies have found that HIV risk behavior among PLWHA was determined by patient age and sex [28, 29], educational status [18, 30], marital status [31], alcohol consumption [18, 28], duration on ART [31], HIV serostatus disclosure [32–34] and perceived stigma [35, 36].

HIV prevention policies and programs have concentrated heavily on HIV-negative individuals.

However, strategies to decrease the infectiousness of PLWHA to prevent secondary HIV transmission should be an integral part of the prevention policy to achieve the United Nations Sustainable Development Goal (UN SDG) 3.3 which aims to end the epidemics of AIDS by 2030 [37, 38]. Bearing in mind the high rate of HIV infection in low-income countries

including Ethiopia, various strategies have been tried so far to decrease high-risk sexual practice among PLWHA, in particular, consistent condom use to prevent further transmission of HIV/AIDS among PLWHA and in the general population [39, 40].

Various studies have been conducted to assess HIV risk behavior and associated factors among PLWHA in Ethiopia [4–9, 19, 41–46]. However, studies had inconsistent findings with the prevalence ranging from 15.8% in the Amhara region in the last twelve months before the survey [7] to 56% in Felege Hiwot Referral Hospital in the last three months preceding the survey [19]. Alcohol use, HIV status non-disclosure, and perceived stigma were factors that merit consideration as ones that could impact sexual behaviors and might be amenable to interventions.

Thus, this meta-analysis aimed to estimate the pooled prevalence of HIV risk behavior and associated factors among PLWHA in Ethiopia. The study generates crucial evidence that will be an input for program planners, policymakers, and health service providers to design and implement evidence-based interventions to reduce the transmission of HIV.

## Materials and method

### Study design and searching strategies

This systematic review and meta-analysis was conducted to estimate the pooled prevalence of HIV risk behavior and associated factors among PLWHA in Ethiopia. This meta-analysis followed the Preferred Reporting Items for Systematic Reviews and Meta-Analysis (PRISMA) checklist [47] (S1 Table). Six international databases—Google Scholar, Pub Med, Cochrane Library, HINARI, CINAHL, and Global Health—were searched systematically to find all relevant articles. A snowball search for the references of relevant articles was also performed. In addition, digital libraries were searched to identify grey literature. The search of research articles was carried out from June 1 to September 30, 2020, by three authors (YD, MY, and BK), and articles published from 2000 up to September 30, 2020, were included in the review.

Endnote software was used to collect, organize, and remove the duplications of search outcomes. All potential articles were accessed by using following keywords: "HIV risk behavior", "high risk behavior", "risky sexual practice", "risky behavior", "sexual behavior", "unprotected sexual practice","condom use", "inconsistent condom use", "consistent condom use", "casual sex", "multiple sexual partner", "alcohol use", "alcohol consumption", "HIV status disclosure", "perceived stigma", "associated factors", "HIV patients", "AIDS patients", "people living with HIV/AIDS", "ART attendees", "HIV positive adults", "Ethiopia" independently and in combination using Boolean operators "OR" or "AND". The search strategy was formulated by using Medical Subject Headings (MESH), adding terms and keywords from a primary search (S1 File).

### Inclusion criteria

**Population.**   This meta-analysis included studies conducted among male, female, or both male and female adults living with HIV/AIDS in Ethiopia.

**Exposure.**   PLWHA who drank alcohol, patients who did not disclose HIV serostatus, and those who experienced perceived stigma.

**Comparison.**   PLWHA who didn't drink alcohol, who disclosed their HIV serostatus, and those who did not experience perceived stigma.

**Outcome.**   Studies assessed HIV risk behavior as a primary outcome, whether same-sex or different-sex risk behavior.

**Study design.**   All types of observational studies (cross-sectional, case-control, and cohort) were included.

**Study setting.** All facility-based, as well as community-based studies
**Time frame.** Articles published from the beginning of 2000 up to September 30, 2020
**Country.** Studies conducted only in Ethiopia
**Language.** Articles written in the English language
**Publication.** Published articles were included in this study.

## Outcome measurement

The primary outcome of this study was the pooled prevalence of HIV risk behavior among PLWHA in Ethiopia, which was computed by dividing the number of PLWHA practicing HIV risk behavior by the total sample size multiplied by 100.

The second outcome was the association between alcohol use, HIV status non-disclosure, and perceived stigma and HIV risk behavior in the form of the log odds ratio.

Studies measured history of alcohol use differently; three studies measured it as consuming any alcoholic beverage in the last three months [4, 42, 46], one study measured it as ever used alcohol [7], and the remaining study measured it as alcohol consumption in the last 12 months [41].

Primary studies measured HIV serostatus disclosure as "YES" if respondents had disclosed their HIV positive status to their recent sexual partner and "NO" if they had not [4, 19, 41–44, 46].

Primary studies calculated perceived stigma from respondents' responses to stigma-specific questions. It was measured as "having perceived stigma" if respondents scored greater than or equal to the mean value and "have no perceived stigma" if respondents scored below the mean value [4, 19, 42, 43].

## Operational definitions

**HIV risk behavior.** Is defined as practicing one or more of the following: sex without the use of condoms, inconsistent condom use, having multiple sexual partners, casual sex, sex with the influence of alcohol, and sexual exchange (exchange of money for sexual intercourse).

**Inconsistent condom use.** Responding "never", "sometimes", and "usually" to questions regarding the frequency of condom use.

**Condom less sex.** Not using a condom all the time for every act of sexual intercourse.

**Multiple sexual partners.** Having more than one sexual partner.

**Casual sex.** Sex with a non-regular sexual partner.

**Alcohol use.** Alcohol use in this study was measured as ever consuming any alcoholic beverage.

## Data extraction

Six reviewers (YD, BA, MA, AM, MG, and WM) independently extracted all the necessary data from the relevant studies using a standardized and a pretested data extraction format prepared in a Microsoft™ Excel spreadsheet. The data extraction format includes the name of the first author followed by initials, region, publication year, study area, study setting (whether it is institution- or community-based), study design, sample size, response rate, the residence of PLWHA, number of PLWHA with HIV risk behavior, prevalence of HIV risk behavior, data collection method, the definition of outcome variable across the included studies, the definition of alcohol use across the included studies and frequencies of alcohol use, HIV status non-disclosure and perceived stigma in the form of two-by-two tables. Disagreement raised at the time of data extraction was resolved through discussion. Whenever possible, the corresponding authors of the research articles were contacted for clarification and additional information.

## Quality assessment

After duplicate files were removed, the relevant articles identified from all databases were screened for inclusion by three reviewers (YD, BK, and MY). Then the Joana Brigg Institute (JBI) critical appraisal checklist for prevalence studies was used to assess the quality of each article (S2 File).

The checklist has nine questions with four response options (yes, no, unclear, and not applicable). These nine questions are: Was the sample frame appropriate to address the target population?, Were study participants sampled appropriately?, Was the sample size adequate?, Were the study subjects and the setting described in detail?, Was the data analysis conducted with sufficient coverage of the identified sample?, Were valid methods used for the identification of the condition?, Was the condition measured in a standard, reliable way for all participants?, Was there appropriate statistical analysis? And was the response rate adequate, and if not, was the low response rate managed appropriately?

Three authors (YD, BA, and MA) independently assessed the quality of each study out of 100%. In the JBI quality assessment tool, 50% was used as a cutoff point for the inclusion of research articles [48, 49]. In this study, since all of the included articles scored 50% or more, all are included in the review. The difference in the result at the time of quality assessment was settled by taking the mean score of the results of all reviewers.

## Data analysis

The relevant data extracted using a Microsoft Excel spreadsheets were exported into STATA/SE version-14 statistical software for analysis. The heterogeneity among the included studies was statistically estimated by using the $I^2$ test, with a variation in outcomes greater than 75% being taken as high heterogeneity.

Due to the presence of severe heterogeneity between the studies, DerSimonian and Liard's method of random effect model at a P-value less than 0.05 with a 95% confidence interval was used to estimate the pooled prevalence of HIV risk behavior in Ethiopia [50]. Subgroup analysis was performed by various study level characteristics such as region (Amhara, Addis Ababa, Oromia, and Southern Nation Nationalities and Peoples' Region [SNNPR]), residence (urban versus urban and rural), sample size (<400 versus ≥400) and the type of outcome assessed among the included articles. Moreover, univariate meta-regression analysis was done using sample size and publication year as factors to identify the possible source of heterogeneity and to decrease the random variations among the point estimates of original articles.

Moreover, a forest plot was used to present the point estimates with their 95% confidence interval, and a log odds ratio was used to determine the association between alcohol use, HIV status non-disclosure, and perceived stigma with HIV risk behavior among PLWHA. Finally, the presence of publication bias was objectively assessed using Egger's regression test at a p-value < 0.05, which was considered a statistically significant publication bias [51].

## Results

### Study selection

A total of four thousand one hundred thirty-seven (4,137) published and unpublished studies were identified through electronic databases (Google Scholar, Cochrane Library, Pub Med, CINAHL, HINARI, and Global Health) and a digital library search. Among these, 4,123 articles were excluded as a result of duplication, due to their titles and abstracts, and as a result of not fulfilling the inclusion criteria. Finally, 14 eligible articles were included for analysis (Fig 1).

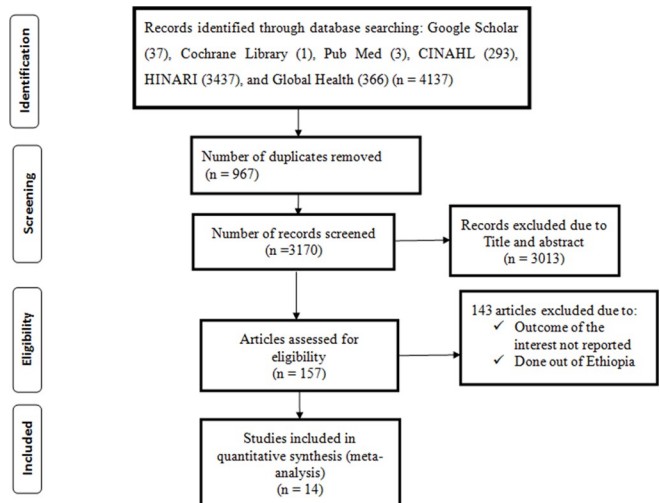

**Fig 1. PRISMA flow diagram describing the selection of studies for systematic review and meta-analysis.**

## Characteristics of the included studies

All the fourteen studies were institution-based cross-sectional studies published between 2008 and 2020 [4–9, 19, 41–46]. A total of 6,179 PLWHA were involved to estimate the pooled prevalence of HIV risk behavior and associated factors among PLWHA in Ethiopia. The sample size of the included studies ranged from a minimum of 109 samples in a study conducted at Felege Hiwot Referral Hospital, Amhara region [19] to a maximum of 745 samples in a study conducted in the same region [7].

The lowest prevalence of HIV risk behavior (15.8%) was reported by a study done in the Amhara region [7] whereas the highest prevalence of HIV risk behavior (56%) was observed by a study done at Felege Hiwot Referral Hospital, Amhara region [19]. The meta-analysis included 7studies from Amhara region [6–9, 19, 44], 3 studies from Oromia region [41, 43, 45], 3 studies from Addis Ababa [4, 5, 42]and 1 study from Southern Nation Nationalities and Peoples Region (SNNPR) [46] (Table 1).

## The prevalence of HIV risk behavior in Ethiopia

Analysis of the results of 14 studies showed that the pooled prevalence of HIV risk behavior among PLWHA in Ethiopia was 34.3% (95% CI: 28.2, 40.3). Severe heterogeneity was observed among the included research articles ($I^2$ = 96.6, p = 0.000). As a result, the DerSimonian and Laird random-effects meta-analysis model was used to estimating the pooled prevalence of HIV risk behavior (**Fig 2**).

The presence of publication bias was subjectively assessed using a funnel plot. The plot showed an asymmetrical distribution of the effect estimates indicating the presence of publication bias (Fig 3). However, Eggers test statistics confirm the absence of significant publication bias (P = 0.107). Furthermore, a sensitivity analysis was done to identify a single study's effect on the overall pooled estimate. As a result, no evidence of a single study effect on the overall pooled prevalence was found.

## Subgroup analysis

To identify the variation among the individual studies, subgroup analysis was conducted based on the region where the studies were done, residence, sample size, and the type of outcome

**Table 1. Descriptive summary of fourteen studies included in the meta-analysis of HIV risk behavior and associated factors among PLWHA in Ethiopia, 2020.**

| Authors | Publication year | Region | Study area | Study design | Sex | Outcome assessed | Definition of alcohol use | Data collection method | Sample size | Response rate | Prevalence (%) | Quality score |
|---|---|---|---|---|---|---|---|---|---|---|---|---|
| Dessie et al.[4] | 2011 | Addis Ababa | Addis Ababa | Cross-sectional | Both | Condom-unprotected sex | Alcohol use in the last three months | Face-to-face interview | 601 | 100 | 36.9 | 85.2% |
| Tadesse and Gelagay [5] | 2019 | Addis Ababa | Addis Ababa | Cross-sectional | Both | Sex without condoms and/or, inconsistent condoms use and/or multiple sexual partners and/or casual sex and/or sex with the influence of alcohol and/or sexual exchange for money | | Face-to-face interview | 562 | 100 | 39.1 | 81.5% |
| Alene et al. [8] | 2014 | Amhara | Gondar, Dessie and DebreMarkos | Cross-sectional | Female | Inconsistent condom use | | Face-to-face interview | 351 | 98 | 43.3 | 63% |
| Engedashet et al. [41] | 2014 | Oromia | Deberezeit | Cross-sectional | Both | Condom-unprotected sex | Alcohol use in the past one year | Face-to-face interview | 667 | 100 | 22.2 | 85.2% |
| Demissie et al.[42] | 2015 | Addis Ababa | Addis Ababa | Cross-sectional | Both | Inconsistent condom use and/or unprotected sex | Alcohol use in the last three months | Face-to-face interview | 376 | 100 | 30.4 | 66.7% |
| Molla and Gelagay [6] | 2017 | Amhara | Gondar | Cross-sectional | Both | Multiple sexual partners and/or casual sex and/or sex without condom and/or inconsistent condom use and/or sex with the influence of alcohol. | | Face-to-face interview | 513 | 99 | 38 | 77.8% |
| Mosisa et al. [43] | 2018 | Oromia | Nekemte | Cross-sectional | Both | Inconsistent use of condoms and/or no condom-protected sex with HIV-negative | | Face-to-face interview | 337 | 100 | 32.9 | 66.7% |
| Ali et al. [44] | 2019 | Amhara | KollaDiba | Cross-sectional | Both | Inconsistent use of condoms | | Face-to-face interview | 358 | 91 | 43.5 | 59.2% |
| Shewamene et al. [9] | 2015 | Amhara | Gondar | Cross-sectional | Both | Inconsistent use of condoms | | Face-to-face interview | 317 | 100 | 21.1 | 66.7% |
| Yalew et al.[19] | 2012 | Amhara | FelegeHiwot | Cross-sectional | Both | Inconsistent use of condoms | | Face-to-face interview | 113 | 100 | 44.2 | 55.6% |
| Yalew et al. [19] | 2012 | Amhara | FelegeHiwot | Cross-sectional | Both | Inconsistent use of condoms | | Face-to-face interview | 109 | 100 | 56 | 55.6% |
| Deribe et al. [45] | 2008 | Oromia | Jimma | Cross-sectional | Both | Unprotected sex | | Face-to-face interview | 705 | 100 | 24 | 88.9% |
| Moges et al. [7] | 2020 | Amhara | Amhara region | Cross-sectional | Both | inconsistent condom use and/or not using a condom | Ever used alcohol | Face-to-face interview | 745 | 98 | 15.8 | 88.9% |
| Abebo et al. [46] | 2019 | SNNPR | ArbaMinich | Cross-sectional | Both | Unprotected sexual practice | Alcohol use in the last three months | Face-to-face interview | 513 | 100 | 52 | 81.5% |

SSNPR: Southern Nations, Nationalities, and Peoples Region

assessed across studies. Even though heterogeneity still existed in the subgroup analysis of all parameters mentioned above, the result indicated that the lowest pooled prevalence of HIV risk behavior was observed by studies conducted in Addis Ababa as compared to studies conducted in the SNNPR, Oromia, and Amhara regions [35.7% (95% CI: 30.9, 40.4)] (Table 2).

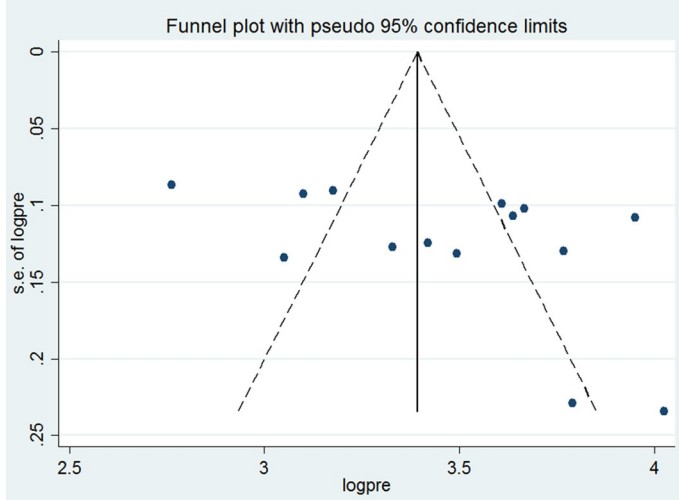

**Fig 2. Forest plot of the pooled prevalence of HIV risk behavior among PLWHA in Ethiopia, 2020.**

In addition, the highest prevalence of HIV risk behavior was observed by studies conducted in both urban and rural settings [35.2% (95% CI: 26.9, 43.5)] and studies with a sample size of ≤400 [35.9% (95% CI: 28.5, 43.4)] as compared to studies conducted only in an urban setting and those with a sample size of >400 (Table 2).

In addition, a univariate meta-regression was conducted using sample size and publication year as factors. However, neither of them was found to be statistically significant sources of heterogeneity. Although not significant sources of variability, as the sample size and publication year increased, the prevalence of HIV risk behavior decreased (Table 3).

## Factors associated with HIV risk behavior

The association between alcohol use and HIV risk behavior was examined based on the result of five studies [4, 7, 41, 42, 46]. The association was positively significant in three studies [4,

**Fig 3. Funnel plot of the pooled prevalence of HIV risk behavior in Ethiopia, 2020.**

**Table 2. Subgroup prevalence of HIV risk behavior in Ethiopia, 2020 (n = 14).**

| Variables | Characteristics | Included studies | Sample size | Prevalence (95% CI) | I$^2$ | P-value |
|---|---|---|---|---|---|---|
| **Region** | Amhara | 7 | 2,506 | 34.8 (24.7, 44.9) | 96.9% | <0.001 |
| | Addis Ababa | 3 | 1,539 | 35.7 (30.9, 40.4) | 74.2% | 0.021 |
| | Others | 4 | 2,222 | 32.7 (19.9, 45.6) | 97.8% | <0.001 |
| **Residence** | Rural and urban | 10 | 4061 | 35.2 (26.9, 43.5) | 97.2% | <0.001 |
| | Urban | 4 | 2,206 | 32.2 (23.9, 40.4) | 94.5% | <0.001 |
| **Sample size** | ≤400 | 7 | 1,961 | 35.9 (28.5, 43.4) | 92.1% | <0.001 |
| | >400 | 7 | 4,306 | 32.5(23.3, 41.7) | 97.9% | <0.001 |
| **Type of outcome assessed** | Unprotected sex | 4 | 2,486 | 33.7 (21.1, 46.3) | 98.0% | <0.001 |
| | Practicing one or more of the following:<br>✓ Sex without condoms<br>✓ Inconsistent condoms use<br>✓ Multiple sexual partners<br>✓ Casual sex<br>✓ Sex with the influence of alcohol<br>✓ Sexual exchange for money | 2 | 1,075 | 38.6 (35.7, 41.5) | 94.2% | <0.001 |
| | Inconsistent condom use | 5 | 1,248 | 38.1 (26.6, 49.6) | 94.7% | <0.001 |
| | Inconsistent condom use or unprotected sex | 3 | 1,458 | 26.3 (14.4, 38.2) | 96.2% | <0.001 |

SNNPR-Southern Nation Nationalities and Peoples Region; Others-Oromia and SNNPR.

41, 46] and not significant in the other two studies [7, 42]. In this meta-analysis, patients who consumed alcohol were 1.9 times more likely to be engaged in HIV risk behavior as compared to their non-alcohol-consuming counterparts (OR = 1.9, 95%, CI: [1.6, 2.3]). A fixed-effect meta-analysis model was used to examine the association between alcohol use and HIV risk behavior due to the absence of significant heterogeneity (I$^2$ = 23.7%, p = 0.264) (Fig 4). Publication bias was also assessed using Egger's tests, the result indicating the absence of publication bias (P = 0.439).

A total of seven studies [4, 19, 41–44, 46] were included to identify the association between HIV status non-disclosure and HIV risk behavior. The association was significant in one study [43] and non-significant in the other studies [4, 19, 41, 42, 44, 46]. The result of the random-effect meta-analysis showed that patients who did not disclose their HIV status to their sexual partner were 2.3 times more likely to be engaged in HIV risk behavior as compared to their counterparts who did disclose their HIV status (OR = 2.3, 95%CI: [1.3, 4.0]) (Fig 5). High heterogeneity was exhibited among the included studies (I$^2$ = 80.4%, p = 0.000) and there was a low possibility of publication bias as indicated by Egger's tests (P = 0.786).

The effect of perceived stigma on HIV risk behavior was assessed based on the results of four research articles [4, 19, 42, 43]. The effect was positively significant in two studies [19, 43] and non-significant in the other two studies [4, 42]. In this study, patients who experienced perceived stigma were 2.3 more likely to practice HIV risk behavior as compared to patients without perceived stigma (OR = 2.3, 95%CI: [1.3, 4.1]) (Fig 6). The random effect meta-analysis model was used to estimate the pooled effect due to the presence of severe heterogeneity

**Table 3. Univariate meta-regression analysis to determine factors related to the heterogeneity of the prevalence of HIV risk behavior in Ethiopia, 2020.**

| Variables | Coefficient | P-value |
|---|---|---|
| Sample size | -0.0336989 | 0.053 |
| Year of publication | 0.2425176 | 0.792 |

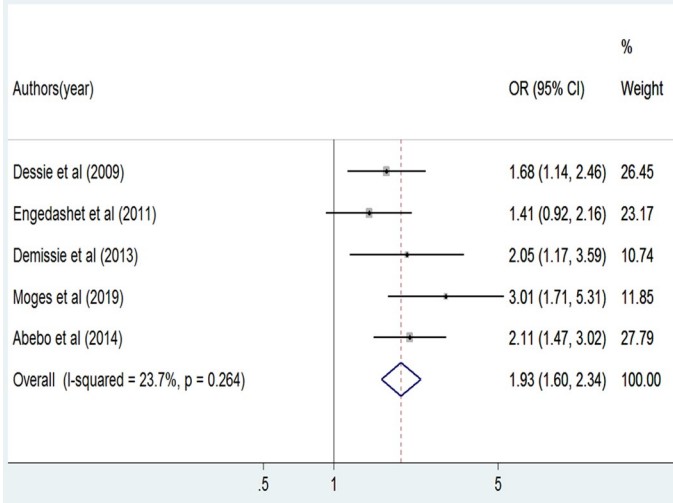

**Fig 4. The pooled odds ratio of alcohol use among PLWHA in Ethiopia, 2020.**

among studies ($I^2$ = 79.2%, p = 0.002) and Egger's tests showed the absence of publication bias (P = 0.682).

## Discussion

In this meta-analysis, the pooled prevalence of HIV risk behavior is congruent with studies conducted in Sao Paulo, Brazil (28.7%) [28], South Africa (30%) [52], Togo (34.6%) [18], Tanzania (40%) [53], and another study conducted in Italy (40%) [54]. However, it is higher than the results found by studies conducted in India (13%) [55], Kenya (28%) [17], Croatia (23%) [56], and South Africa (24.2%) [30].

Similarly, the finding is higher than that of other studies conducted in the United States (23%) [57] and Jamaica (25%) [32]. But the finding was lower than studies conducted in Nigeria (56%) [58], Kumasi, Ghana (51%) [59], and Uganda (58.7%) [16] and another study

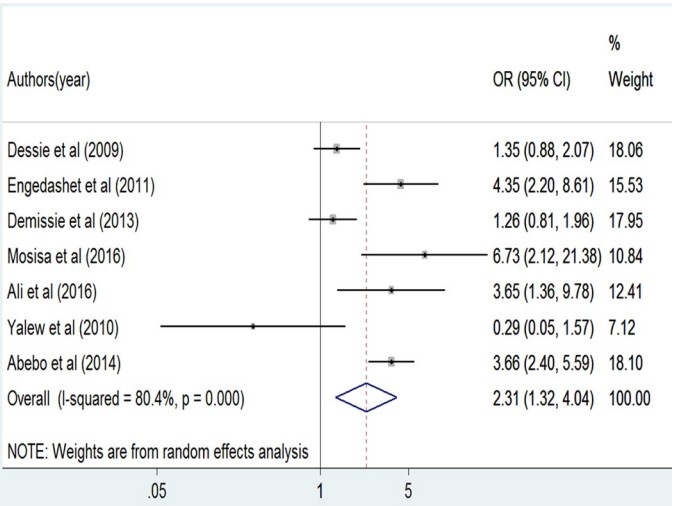

**Fig 5. The pooled odds ratio of HIV status non-disclosure among PLWHA in Ethiopia, 2020.**

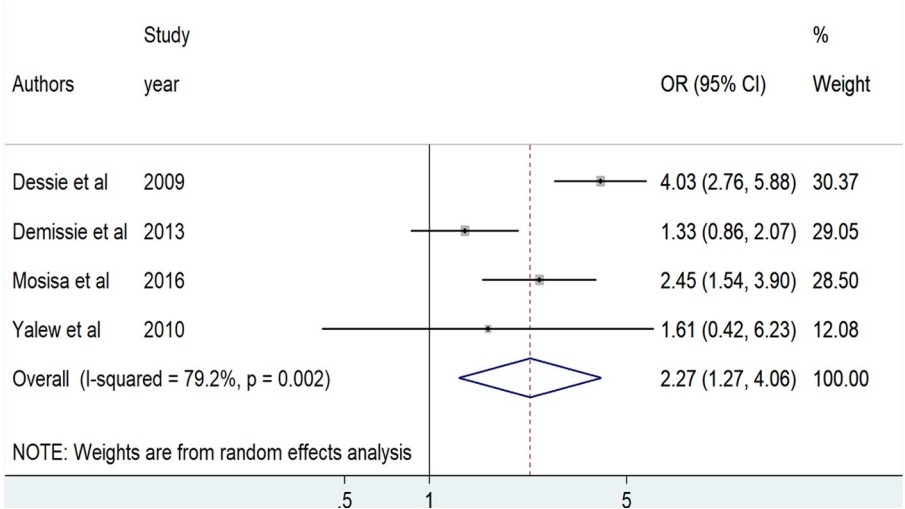

**Fig 6. The pooled odds ratio of perceived stigma among PLWHA in Ethiopia, 2020.**

conducted in Nigeria (42.4%) [29]. The variation could be due to differences in the study settings, the differences in their definition used to define HIV risk behavior, time variation, and the difference in socio-economic status, educational status, and cultural and contextual factors.

The other possible causes for the variation are the differences in the length of time used to measure the prevalence of HIV risk behavior, methodological variation (data collection method used and sampling of study participants), and the difference in the concern of different governmental and non-governmental organizations in HIV prevention and control across countries. For example, all studies included in this analysis used face-to-face interviews to collect the data and most of them assessed inconsistent condom use as HIV risk behavior [8, 9, 19, 42–44].

The high rate of high-risk sexual behavior among PLWHA in this meta-analysis has implications for the continued risk of HIV transmission in the country despite the increasing access to ART treatment. Although ART treatment reduces the risk of HIV transmission [60, 61], the finding of this study calls for designing behavioral intervention programs while at the same time scaling up access to Highly Active Antiretroviral Treatment (HAART) to address high-risk sexual behavior and to reduce HIV transmission.

The result of subgroup analysis indicated that the lowest pooled prevalence of HIV risk behavior was observed by studies conducted in Addis Ababa as compared to studies conducted in SNNPR, Oromia, and Amhara regions. This could be due to differences in educational status, media exposure, and difference in knowledge of HIV prevention methods and comprehensive knowledge about HIV/AIDS. For instance, according to Ethiopian Demographic and Health Survey (EDHS) 2016, respondents who lived in Addis Ababa had better knowledge of HIV prevention methods and comprehensive knowledge about HIV/AIDS than respondents who lived in SNNPR, Oromia, and Amhara regions [62].

The likelihood of engagement in HIV risk behavior was higher among PLWHA who consumed alcohol as compared to their non-drinking counterparts. This is in agreement with studies conducted in South Western Uganda [63], South Africa [52], and New Guinea [64]. The results of studies conducted in Sao Paulo, Brazil [28], Togo [18], and another study conducted in Uganda [16] also showed that alcohol consumption had a positive association with HIV risk behavior.

The finding of a systematic review and meta-analysis conducted in the United States also witnessed this association [65]. This association may be because alcohol use can inhibit an individual's perception of the risk of HIV transmission and hinder thinking and decision-making ability about safe sex as a result of alcohol's restricting effect on cognitive capacity causing someone to focus only on impelling immediate cues [66]. Bearing in mind the strong association between alcohol use and HIV risk behavior, clinicians need to integrate alcohol counseling into routine HIV care for PLWHA. It is also necessary to implement policies and strategies focusing on decreasing alcohol consumption and HIV risk behavior among PLWHA.

Non-disclosure of HIV serostatus to sexual partners had a significant association with HIV risk behavior. This is consistent with studies conducted in Jamaica [32], Cape Town, South Africa [33], and Cameroon [34] where HIV risk behavior was reported mostly among patients who did not disclose their HIV status to sex partners. A similar finding was also documented in a study conducted in Johannesburg, South Africa [30]. This could be due to the possibility that PLWHA who did not know the serostatus of their sexual partners' might not be driven to use a condom at the time of sexual intercourse.

Disclosing HIV status is the key component of HIV prevention as it encourages partners to know each other's HIV status, increases adherence to therapy, and improves the rate of CD4 recovery following ART treatment. Studies showed that HIV status disclosure decreases the risk of HIV transmission by 17.9% to 40.6% [67, 68]. However, a recent systematic review and meta-analysis study conducted in Ethiopia showed that 25.6% of HIV patients did not disclose their HIV status to their sexual partners [69]. More health education interventions including extensive adherence counseling of all PLWHA about the benefit of the disclosure are required to increase the HIV status disclosure rate and to decrease high-risk sexual behavior among PLWHA.

Experiencing perceived stigma also had a significant effect on HIV risk behavior. The finding is in line with studies conducted in India and the United States [35, 36]. This might be due to the possibility that individuals who have experienced perceived stigmatization could be afraid to disclose their HIV serostatus to their sexual partners. In Ethiopia, HIV-related stigma is the most common problem affecting 42–72.2% of HIV patients despite the effort made by NGOs and the government [70, 71]. Stigma not only affects the sexual behavior of HIV patients but also hinders ART adherence and thereby hastens disease progression. Thus, the government and all other concerned bodies should be a voice for PLWHA and do more to confront and reduce HIV-related stigma.

Like other studies, the results of this meta-analysis must be interpreted carefully in accordance with the strengths and limitations of the included studies. As far as the authors know, this systematic review and meta-analysis was the first study that attempts to assess the pooled prevalence of HIV risk behavior among PLWHA in the Ethiopian context, although it has some limitations. As a limitation, it is restricted to articles published in the English language and it may not represent research published in other languages.

The approaches used to measure independent and outcome variables differ across the included studies. Besides, all of the articles included in the analysis were cross-sectional and had a small sample size and this may have affected the pooled estimate. In addition, heterogeneity was detected across all analyses even if we conducted subgroup analyses and meta-regression. Moreover, the study may not be representative of all regions since the included articles were only from one city administration and three regions of Ethiopia.

## Conclusions

The pooled prevalence of HIV risk behavior among PLWHA in Ethiopia was high. Alcohol use, HIV status non-disclosure, and perceived stigma were factors significantly associated with

HIV risk behavior. Besides increasing access and adherence to ART treatment, the national program against HIV/AIDS should enhance health promotion activities including consistent condom use and the creation of a positive social environment to mitigate high-risk sexual practices among PLWHA. Screening and treatment of individuals with alcohol use disorder should be an integral part of the routine HIV treatment, care, and support package and health care providers should offer risk reduction counseling, especially for clients who use alcohol and/or experience perceived stigma, including encouraging disclosure of HIV serostatus among patients whose sexual partners are of unknown status.

## Supporting information

**S1 Table. Preferred Reporting Items for Systematic Reviews and Meta-Analysis (PRISMA) checklist.**
(DOC)

**S1 File. Search strategy used to estimate the pooled prevalence of HIV risk behavior and associated factors among PLWHA in Ethiopia.**
(DOCX)

**S2 File. JBI critical appraisal checklist for prevalence studies.**
(DOCX)

**S1 Dataset. The data set used to estimate the pooled prevalence of HIV risk behavior and associated factors among PLWHA in Ethiopia.**
(XLSX)

## Acknowledgments

The authors gratefully acknowledge the Wollo University for making available its library online database.

## Author Contributions

**Conceptualization:** Yitayish Damtie, Metadel Adane.

**Data curation:** Yitayish Damtie, Metadel Adane.

**Formal analysis:** Yitayish Damtie, Metadel Adane.

**Investigation:** Yitayish Damtie, Bereket Kefale, Melaku Yalew, Mastewal Arefaynie, Bezawit Adane, Amare Muche, Reta Dewau, Zinabu Fentaw, Erkihun Tadesse Amsalu, Gedamnesh Bitew, Wolde Melese Ayele, Assefa Andargie Kassa, Muluken Genetu Chanie, Mequannent Sharew Melaku, Metadel Adane.

**Methodology:** Yitayish Damtie, Bereket Kefale, Melaku Yalew, Mastewal Arefaynie, Bezawit Adane, Amare Muche, Reta Dewau, Zinabu Fentaw, Erkihun Tadesse Amsalu, Gedamnesh Bitew, Wolde Melese Ayele, Assefa Andargie Kassa, Muluken Genetu Chanie, Mequannent Sharew Melaku, Metadel Adane.

**Project administration:** Yitayish Damtie, Bereket Kefale, Melaku Yalew, Mastewal Arefaynie, Bezawit Adane, Amare Muche, Reta Dewau, Zinabu Fentaw, Erkihun Tadesse Amsalu, Gedamnesh Bitew, Wolde Melese Ayele, Assefa Andargie Kassa, Muluken Genetu Chanie, Mequannent Sharew Melaku, Metadel Adane.

**Resources:** Yitayish Damtie, Bereket Kefale, Melaku Yalew, Mastewal Arefaynie, Bezawit Adane, Amare Muche, Reta Dewau, Zinabu Fentaw, Erkihun Tadesse Amsalu, Gedamnesh Bitew, Wolde Melese Ayele, Assefa Andargie Kassa, Muluken Genetu Chanie, Mequannent Sharew Melaku, Metadel Adane.

**Software:** Yitayish Damtie, Bereket Kefale, Melaku Yalew, Mastewal Arefaynie, Bezawit Adane, Amare Muche, Reta Dewau, Zinabu Fentaw, Erkihun Tadesse Amsalu, Gedamnesh Bitew, Wolde Melese Ayele, Assefa Andargie Kassa, Muluken Genetu Chanie, Mequannent Sharew Melaku, Metadel Adane.

**Supervision:** Yitayish Damtie, Bereket Kefale, Melaku Yalew, Mastewal Arefaynie, Bezawit Adane, Amare Muche, Reta Dewau, Zinabu Fentaw, Erkihun Tadesse Amsalu, Gedamnesh Bitew, Wolde Melese Ayele, Assefa Andargie Kassa, Muluken Genetu Chanie, Mequannent Sharew Melaku, Metadel Adane.

**Validation:** Yitayish Damtie, Bereket Kefale, Melaku Yalew, Mastewal Arefaynie, Bezawit Adane, Amare Muche, Reta Dewau, Zinabu Fentaw, Erkihun Tadesse Amsalu, Gedamnesh Bitew, Wolde Melese Ayele, Assefa Andargie Kassa, Muluken Genetu Chanie, Mequannent Sharew Melaku, Metadel Adane.

**Visualization:** Yitayish Damtie, Bereket Kefale, Melaku Yalew, Mastewal Arefaynie, Bezawit Adane, Amare Muche, Reta Dewau, Zinabu Fentaw, Erkihun Tadesse Amsalu, Gedamnesh Bitew, Wolde Melese Ayele, Assefa Andargie Kassa, Muluken Genetu Chanie, Mequannent Sharew Melaku, Metadel Adane.

**Writing – original draft:** Yitayish Damtie, Metadel Adane.

**Writing – review & editing:** Yitayish Damtie, Metadel Adane.

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
