## [Decision Letter · Decision Letter 0]

17 Nov 2021

PONE-D-21-17808

Risky sexual practice and its association with alcohol intake, HIV status disclosure, and perceived stigma among adult HIV infected patients in Ethiopia. A systematic review and meta-analysis

PLOS ONE

Dear Dr. Adane,

Thank you for submitting your manuscript to PLOS ONE. After careful consideration, we feel that it has merit but does not fully meet PLOS ONE’s publication criteria as it currently stands. Therefore, we invite you to submit a revised version of the manuscript that addresses the points raised during the review process.

I agree with the Reviewers' comments and addressing these will be a requirement for acceptance. In addition, the authors must revisit the concept of "risky" sexual behavior prior to acceptance. People living with HIV who use antiretroviral therapy and attain a suppressed viral load cannot transmit HIV through sexual contact, and this is a critical concept that is not highlighted in the article, and which must be integrated into any study on sexual behaviors among people living with HIV (or those who have sex with members of this population). Further, the term "risky" is stigmatizing, and so it is no longer appropriate for describing sexual behaviors. The term "HIV risk behaviors" is more acceptable. The authors have also not adequately discussed the reasons why people engage in what they define as "risky" behaviors; is it for sexual pleasure, because of alcohol use disorders, or other reasons that can help inform ways to improve HIV prevention intervention in Ethiopia? Finally, I agree that the services of a copy editing expert and more detailed proofreading will be essential to ensure that the paper is easy to read and suitable for publication in this journal.

We look forward to receiving your revised manuscript.

Kind regards,

Douglas S. Krakower, MD

Academic Editor

PLOS ONE

Journal Requirements:

Reviewers' comments:

Reviewer's Responses to Questions

**Comments to the Author**

1. Is the manuscript technically sound, and do the data support the conclusions?

Reviewer #1: Yes

Reviewer #2: Partly

2. Has the statistical analysis been performed appropriately and rigorously? 

Reviewer #1: Yes

Reviewer #2: I Don't Know

3. Have the authors made all data underlying the findings in their manuscript fully available?

Reviewer #1: Yes

Reviewer #2: Yes

4. Is the manuscript presented in an intelligible fashion and written in standard English?

Reviewer #1: Yes

Reviewer #2: No

5. Review Comments to the Author

Reviewer #1: Title: change “adult HIV infected patients” to “adults living with HIV”

Key words: revise to remove standard terms i.e. prevalence.

Terminology: use correct terminology for “HIV infected patients”; use consistently: PLWHA

Abstract

Line 23: risky sexual practice does not increase HIV incidence; it increases the risk of HIV transmission.

Background

Line 53: CDC only used once so no need for abbreviation.

Line 54: lower case for “Sexually Transmitted Infection”. One does not contract AIDS but HIV, please edit the sentence.

Line 60: Use abbreviation PLWHA

Line 80-83: Consider revising or add to methods or discussion.

Methodology

Line 118-124: What was the inclusion criteria? The section on inclusion criteria is not clear. Were studies included limited to those conducted in Ethiopia?

Line 131-133: revise sentence e.g. Studies measured alcohol intake as “taking any alcoholic beverage in the last three months” [24], one study used “ever used alcohol” [35] and the remaining one study measured it as “alcohol consumption in the last 12 months” [27]

Line 135: Risky sexual practice?

Line 140: Six reviewers….

Line 141: reference the pretested data extraction format.

Line 147-149: Were there any corresponding authors that were contacted for clarification?

Line 192-194: Were there any regional differences worth discussing?

Results

Line 197: typo: Severe heterogeneity….

Line 209-210: Revise the first two sentences. Was this meant to be a standalone paragraph?

Line 216-217: Please add a comment in your discussion regarding regions.

Line 222: Use scientific writing… What does “a little bit high” mean? Please revise.

Line 229: For OR, please use 1 decimal point throughout and be consistent.

Discussion

Do not repeat your results in your discussion. You can still highlight specifics but limit repetition of results.

Line 253-254: Is 39% prevalence high? Significant?

Reviewer #2: This a meta-analysis of the prevalence of sexual risk behavior among people living with HIV in Ethiopia. The researchers also examined the relationship between sexual risk behavior and alcohol use, stigma, and HIV disclosure. Although the study covers an important topic, it has some conceptual and methodological problems.

The researchers justify the study by indicating that it will solve controversies regarding the factors that affect sexual risk behaviors in Ethiopia. However, there is no discussion of the differences the study intends to address. Moreover, although alcohol use and stigma may affect risk behaviors, moderators such as such as age, gender, access to treatment, and type of partner are likely to explain the heterogeneity in the reviewed studies. In fact, the characteristics and behaviors of the sample included in the review are not described.

The description of the outcome and predictor variables is incomplete. Risk behaviors include number of sexual partners, inconsistent condom use, and having had casual partners. How were these standardized? What happened with studies that included multiple risk-behavior measures or reported continuous outcomes (e.g., average number of sexual partners)? Given the high heterogeneity in the studies and the diverse indicators of risk behaviors, it would be important to conduct moderator analyses in terms of type of outcome measure, in addition to key demographic characteristics aside from the study setting.

How were stigma and HIV status disclosure measured in the primary studies? Were the variables dichotomous?

The description of the quality assessment is incomplete, how many studies were excluded because of the quality assessment?

In the discussion, the extensive descriptions of the levels of risk behavior in different regions needs to be streamlined so it is easier to compare and interpret findings.

It is difficult to conclude that alcohol use affects sexual risk behaviors, given the measures that are included in the studies (e.g., lifetime alcohol use, alcohol use last 12 months).

I would recommend having the paper revised by a native English speaker or professional translator. There are lines that sound awkward or are unclear, and/or need to be revised in terms of word choice and punctuation. There is also some inconsistency in the use of terms. In addition, some paragraphs would benefit from reorganization.

6. PLOS authors have the option to publish the peer review history of their article (what does this mean?). If published, this will include your full peer review and any attached files.

Reviewer #1: **Yes: **Tafireyi Marukutira

Reviewer #2: No

---

## [Author Response · Author response to Decision Letter 0]

29 Dec 2021

Douglas S. Krakower, MD

RE: Manuscript ID: PONE-D-21-17808 (HIV risk behavior and its association with alcohol intake, HIV status non-disclosure, and perceived stigma among people living with HIV/AIDS in Ethiopia: A systematic review and meta-analysis.)

Dear Dr. Douglas S. Krakower,

Thank you very much for your email and the comments/suggestions of the reviewers and academic editor. We have looked at the comments and have revised our paper accordingly. We hope our paper improved as a result of incorporating the reviewers' and academic editor's comments and suggestions.

Please find for your kind consideration the following:

A rebuttal letter that responds to each point raised by the academic editor and reviewer labeled 'Response to Reviewers'. The point-by-point responses are written in italic font style.

A revised manuscript with track changes labeled 'Revised Manuscript with Track Changes'.

A revised paper without tracked changes labeled 'Manuscript'

While hoping that these changes would meet with your favourable consideration, we are happy to hear if there are more comments and suggestions. Please do not hesitate to let us know if you have any questions.

Yours Sincerely,

Metadel Adane

Department of Environmental Health, Wollo University 

Dessie, Ethiopia

Tel:+251910336962

E-mail: metadel.adane2@gmail.com

Point by point response

Editor Comments:

I agree with the Reviewers' comments and addressing these will be a requirement for acceptance. In addition, the authors must revisit the concept of "risky" sexual behavior prior to acceptance. People living with HIV who use antiretroviral therapy and attain a suppressed viral load cannot transmit HIV through sexual contact, and this is a critical concept that is not highlighted in the article, and which must be integrated into any study on sexual behaviors among people living with HIV (or those who have sex with members of this population). Further, the term "risky" is stigmatizing, and so it is no longer appropriate for describing sexual behaviors. The term "HIV risk behaviors" is more acceptable. The authors have also not adequately discussed the reasons why people engage in what they define as "risky" behaviors; is it for sexual pleasure, because of alcohol use disorders, or other reasons that can help inform ways to improve HIV prevention intervention in Ethiopia? Finally, I agree that the services of a copy editing expert and more detailed proofreading will be essential to ensure that the paper is easy to read and suitable for publication in this journal.

Response: Thank you for your constructive comment. We have tried to highlight this critical concept in the introduction section line 58-67. In addition, we have tried to use the term “HIV risk behavior” throughout the document. Furthermore, efforts were made to revise our document by language expert. 

Reviewer #1: 

1. Title: change “adult HIV infected patients” to “adults living with HIV”

Response: Thank you dear reviewer. We have amended it.

2. Key words: revise to remove standard terms i.e. prevalence.

Response: Thank you. We have removed it.

3. Terminology: use correct terminology for “HIV infected patients”; use consistently: PLWHA

Response: Thank you for your comment. We have tried to use the term “PLWHA” throughout our document accordingly. 

Abstract

4. Line 23: risky sexual practice does not increase HIV incidence; it increases the risk of HIV transmission.

Response: Thank you very much for your constructive comments. We have tried to modify it accordingly. 

Background

5. Line 53: CDC only used once so no need for abbreviation.

Response: Thank you for your suggestion. We have amended it. 

6. Line 54: lower case for “Sexually Transmitted Infection”. One does not contract AIDS but HIV, please edit the sentence.

Response: Thank you. We have modified it accordingly.

7. Line 60: Use abbreviation PLWHA

Response: Thank you. The comment is accepted and addressed accordingly. 

8. Line 80-83: Consider revising or add to methods or discussion. 

Response: Thank you. We have revised it.

Methodology

9. Line 118-124: What was the inclusion criteria? The section on inclusion criteria is not clear. Were studies included limited to those conducted in Ethiopia?

Response: Many thanks for your comment. It is editorial problem. Studies included in this meta-analysis are limited to those conducted in Ethiopia. These are additional inclusion criteria’s. 

Population: This meta-analysis include studies conducted among male, female or both male and female adults living with HIV in Ethiopia. 

Exposure: PLWHA who took alcohol, disclosed their HIV sero-status and those who experienced perceived stigma. 

Comparison: PLWHA who didn’t take alcohol, patients with HIV sero-status non-disclosure and those without perceived stigma. 

Outcome: Studies assessed risky sexual practice as a primary outcome; whether it is same-sex or different-sex risk behavior.

Study design: all types of observational studies (Cross-sectional, case-control, and cohort) were included.

Study setting: all facility-based, as well as community-based studies

Time frame: articles published from the beginning of 2000 up to September 30, 2020

Country: studies conducted only in Ethiopia 

Language: articles written in the English language 

Publication: both published and unpublished articles were included in this study. 

10. Line 131-133: revise sentence e.g. Studies measured alcohol intake as “taking any alcoholic beverage in the last three months” [24], one study used “ever used alcohol” [35] and the remaining one study measured it as “alcohol consumption in the last 12 months” [27]

Response: Thank you. We have tried to revise it accordingly. 

11. Line 135: Risky sexual practice?

Response: Definitely, it is to mean risky sexual practice. 

12. Line 140: Six reviewers….

Response: Thank you for your suggestion. We have corrected it. 

13. Line 141: reference the pretested data extraction format.

Response: Thank you for your comment. But the comment is not clear for us. How can we reference it? If you want to see the format, the dataset is already uploaded as a supporting information so that you can see the data extraction form from the dataset. If this is not the case, make it clear for the next revision so that we will entertain it accordingly.

14. Line 147-149: Were there any corresponding authors that were contacted for clarification?

Response: Yes.

15. Line 192-194: Were there any regional differences worth discussing?

Response: Thank you for your valuable comment. Although it was not significant, the pooled prevalence of HIV risk behavior was relatively low among studies conducted in Addis Ababa as compared to studies conducted in SNNPR, Oromia and Amhara region. 

Results

16. Line 197: typo: Severe heterogeneity….

Response: Thank you. We have amended it. 

17. Line 209-210: Revise the first two sentences. Was this meant to be a standalone paragraph?

Response: Thank you for your suggestion. We have tried to revise it. 

18. Line 216-217: Please add a comment in your discussion regarding regions.

Response: Thank you. We have tried to discuss the subgroup prevalence of HIV risk behavior across regions accordingly in line 235-237. 

19. Line 222: Use scientific writing… What does “a little bit high” mean? Please revise.

Response: Thank you for your constructive comment. We have tried to revise it accordingly. 

20. Line 229: For OR, please use 1 decimal point throughout and be consistent.

Response: Thank you. The comment is accepted and addressed accordingly. 

Discussion

21. Do not repeat your results in your discussion. You can still highlight specifics but limit repetition of results.

Response: Thank you. We have amended it. 

22. Line 253-254: Is 39% prevalence high? Significant?

Response: Yes. It is high as compared to different studies conducted before. 

Reviewer #2: 

1. This a meta-analysis of the prevalence of sexual risk behavior among people living with HIV in Ethiopia. The researchers also examined the relationship between sexual risk behavior and alcohol use, stigma, and HIV disclosure. Although the study covers an important topic, it has some conceptual and methodological problems.

Response: Thank you. We have tried to modify our document accordingly. 

2. The researchers justify the study by indicating that it will solve controversies regarding the factors that affect sexual risk behaviors in Ethiopia. However, there is no discussion of the differences the study intends to address. Moreover, although alcohol use and stigma may affect risk behaviors, moderators such as such as age, gender, access to treatment, and type of partner are likely to explain the heterogeneity in the reviewed studies. In fact, the characteristics and behaviors of the sample included in the review are not described.

Response: Thank you for your important comment. We have tried to discuss the differences in the result of the primary studies regarding the factors that affect HIV risk behaviors in result section line 244-245, 253-254 and 262-263. Moreover, we have tried to conduct moderator analysis based on the sex of study participants. However, it is difficult to conduct moderator analysis based the patient age, access to treatment since all are on ART and type of sexual partner since the sexual partner is not specified on primary studies. However, the characteristics and behaviors of the samples included in the review are described on Table 1.

3. The description of the outcome and predictor variables is incomplete. Risk behaviors include number of sexual partners, inconsistent condom use, and having had casual partners. How were these standardized? What happened with studies that included multiple risk-behavior measures or reported continuous outcomes (e.g., average number of sexual partners)? Given the high heterogeneity in the studies and the diverse indicators of risk behaviors, it would be important to conduct moderator analyses in terms of type of outcome measure, in addition to key demographic characteristics aside from the study setting.

Response: Thank you for your important comment. We have tried to write the description of the outcome and predictor variables in method section line 135-152. In addition, we have conducted moderator analyses in terms of type of outcome measure and the sex of study participants as indicated in Table 3. But, we did not attempt to standardize definitions of number of sexual partners, inconsistent condom use, and having had casual partners rather we used them as used in each of the articles selected. None of the articles included in the review reported continuous outcomes. 

4. How were stigma and HIV status disclosure measured in the primary studies? Were the variables dichotomous?

Response: Thank you for your constructive comment. Both stigma and HIV status disclosure were dichotomous variables. Primary studies measured HIV sero-status disclosure as “YES” if respondents told their HIV positive status to their recent sexual partner and “NO” if they didn’t. Primary studies calculated perceived stigma from respondents’ response to stigma-specific questions. It was measured as “having perceived stigma” if respondents scored greater than or equal to the mean value and “have no perceived stigma” if respondents scored below the mean value.

5. The description of the quality assessment is incomplete, how many studies were excluded because of the quality assessment?

Response: Thank you for your important comment. We have tried to revise the quality assessment section. In our study, all of the included articles scored 50% and more, thus all are included in the review. 

6. In the discussion, the extensive descriptions of the levels of risk behavior in different regions needs to be streamlined so it is easier to compare and interpret findings.

Response: Thank you for your comment. We have tried to discuss the levels of risk behavior across regions in the discussion section line 286-292

7. It is difficult to conclude that alcohol use affects sexual risk behaviors, given the measures that are included in the studies (e.g., lifetime alcohol use, alcohol use last 12 months).

Response: Thank you for your constructive comment. It is the amount of alcohol used (rather than the timing of alcohol use) that largely determine the sexual behaviour of PLWHA. Since the amount of alcohol determine the HIV risk behaviour of PLWHA, It is also difficult to conclude that alcohol use in the last 3 months affects HIV risk behavior. In our opinion, it is PLWHA who have ever used alcohol and those used alcohol in the last 12 months who are likely to adapt and maintain the behaviour (alcohol use) and have a chance to be addicted with alcohol and hence practice sexual risk behaviour as compared to those who used alcohol recently. Due to these reason we can conclude that lifetime alcohol use, alcohol use last 12 months can affect sexual risk behaviors.

8. I would recommend having the paper revised by a native English speaker or professional translator. There are lines that sound awkward or are unclear, and/or need to be revised in terms of word choice and punctuation. There is also some inconsistency in the use of terms. In addition, some paragraphs would benefit from reorganization.

Response: The whole parts of the manuscript was revised by language expert for typing, formatting, word choice and punctuation issues. We also have tried to reorganize paragraphs and use terms consistently.

---

## [Decision Letter · Decision Letter 1]

1 Apr 2022

PONE-D-21-17808R1HIV risk behavior and its association with alcohol intake, HIV status non-disclosure, and perceived stigma among people living with HIV/AIDS in Ethiopia: A systematic review and meta-analysisPLOS ONE

Dear Dr. Adane,

Thank you for submitting your manuscript to PLOS ONE. After careful consideration, we feel that it has merit but does not fully meet PLOS ONE’s publication criteria as it currently stands. Therefore, we invite you to submit a revised version of the manuscript that addresses the points raised during the review process.

The following will be required for the manuscript to meet criteria for acceptance: 1) Addressing the most recent reviewers' comments that relate to the statistical methods used and their interpretation and discussion, to ensure methodologic rigor in the final published manuscript2) The term "HIV risk behavior" is currently used in an imprecise manner in the paper. Please be specific in the behaviors that you are describing. If you mean condomless sex, or sex with multiple partners or some other behavior, please mention these in specific. "Unprotected" sex is not specific, as PLWH who have a suppressed viral load cannot transmit HIV to their sexual partners, so sex without condoms in this scenario is "protected" from HIV transmission (even if not from other STIs or pregnancy); please change to "condomless sex" if that it the intended meaning. 3) In line 60, the term "premise" is not appropriate, as this suggests that the assertion that follows is uncertain or untested. Please change to: "There is robust evidence that PLWH who are undetectable on ART cannot transmit HIV to their sexual partners." 4) Lines 61 to 67 suggest that acquisition of drug-resistant HIV strains among PLWH is a major factor in the need for second line ART. The totality of the literature does not support this assertion. Further, the references cited are generally more than a decade old, which ignores a much more robust set of studies from 2010 onward that do not support the authors' statements. Without a more evidence-based and updated argument, the introduction is not appropriate for publication. It would be appropriate to state that "However, PLWH may have challenges in accessing and adhering to ART, which can result in detectable viral loads and the potential for HIV transmission" with the support of new references. 5) Lines 80-82: The authors do not justify why the factors listed are the "most fundamental to intervene," and this strong assertion need to be moderated. It would be more appropriate to state that these factors merit consideration as ones that could impact sexual behaviors and might be amenable to interventions.6) Line 129: the authors need to justify why unpublished studies were included in the analyses.7) Lines 136-139: The authors need to state clearly how they operationalized alcohol use given the heterogeneous definitions used in prior studies. Because the definition for this and other covariates of interest, such as HIV status disclosure, were heterogeneous, this needs to be discussed as a major limitation when considering the study findings in the Discussion. 8) In the Discussion, as above, the authors need to be far more precise when discussing HIV risk behaviors, and must specify exactly those behaviors that are being addressed, as opposed to a general term of "HIV risk behaviors," which can mean many different things. Unless all of the studies in lines 278-285 addressed the same behaviors, then it is not appropriate to make quantitative comparisons across studies as the authors have done.9) Line 39 and elsewhere: the term "safe sex" is outdated, non-specific, and not appropriate. Please remove this and specify exactly what is meant, such as how alcohol could influence decisions around condom use, status disclosure or other factors that can affect HIV transmission. 10) Line 347: The term "alcohol abuse" is stigmatizing; please change to "alcohol use" or "alcohol use disorders" or similar. as above, please also remove "unsafe" as a term. 11) The Discussion and Conclusion are missing a critical emphasis on the importance of promoting access and adherence to ART as a way to decrease HIV transmission, as suppressed viral loads are the most effective way to prevent transmission - even more than condoms or any social-behavioral interventions. The authors can discuss how counseling about alcohol, disclosure and stigma are important to decrease HIV transmission among those without use of ART, but ART needs to be mentioned front and center given the immense strength of evidence behind this biomedical strategy. ==============================

We look forward to receiving your revised manuscript.

Kind regards,

Douglas S. Krakower, MD

Academic Editor

PLOS ONE

Reviewers' comments:

Reviewer's Responses to Questions

**Comments to the Author**

1. If the authors have adequately addressed your comments raised in a previous round of review and you feel that this manuscript is now acceptable for publication, you may indicate that here to bypass the “Comments to the Author” section, enter your conflict of interest statement in the “Confidential to Editor” section, and submit your "Accept" recommendation.

Reviewer #3: All comments have been addressed

Reviewer #4: (No Response)

2. Is the manuscript technically sound, and do the data support the conclusions?

Reviewer #3: Yes

Reviewer #4: Partly

3. Has the statistical analysis been performed appropriately and rigorously? 

Reviewer #3: Yes

Reviewer #4: Yes

4. Have the authors made all data underlying the findings in their manuscript fully available?

Reviewer #3: Yes

Reviewer #4: Yes

5. Is the manuscript presented in an intelligible fashion and written in standard English?

Reviewer #3: Yes

Reviewer #4: No

6. Review Comments to the Author

Reviewer #3: I read with great interest the Manuscript titled "Risky sexual practice and its association with alcohol intake, HIV status disclosure, and perceived stigma among adult HIV infected patients in Ethiopia. A systematic review and meta-analysis" which falls within the aim of PLOSE ONE Journal.

In my honest opinion, the topic is interesting enough to attract the readers' attention. Methodology is accurate and conclusions are supported by the data analysis. Nevertheless, authors should clarify some points and improve about operational definition of some variables.

• The rationale that the authors sated is not consistent across the document. For instance, in the abstract the authors said “ this meta-analysis aimed to estimate the pooled prevalence of risky sexual practice and its association with……” but in the background section page 4, line 80-81 the authors said “ However, we only examine the association between risky sexual behavior and alcohol intake, HIV status disclosure, and perceived stigma since they were the most frequently mentioned factors and have controversial findings among the included studies”.

• How the authors operationalized alcohol consumption, perceived stigma, and HIV status disclosure is not clear. Their definition across the studies might be different and authors should add one table concerning the definition of these variables in each studies include din this review.

• In the abstract section, the authors said “ In Ethiopia, findings regarding risky sexual practice have been inconsistent and inconclusive”. However, they said nothing about this in the background section. Please, show us the inconsistence you found by elaborating this section.

Reviewer #4: This is a good manuscript, clearly written, with the correct methodology and correct statistics for meta-analysis. However, it needs revision to be improved.

Tile: It was too long!

"Its association with alcohol intake, HIV status non-disclosure, and perceived stigma" This listed associated factor can be written in a concise manner.

Abstract:

1. International databases such as Google Scholar, the Cochrane Library, HINARI, PubMed, CINAHL, and Global Health. good to identify which one is a data base and which one is a search engine? Furthermore, why are you using HINARI and Global Health? It is good to consider if you consider others, i.e., AJOL, WorldCat

2. Why did you use the word "prevalence" for "HIV risk behavior"?

Background

1. "There is also a premise that PLWHA who achieve and maintain viral load suppression cannot transmit HIV to their HIV-negative sexual partners [11]." I suggested instead of maintaining viral load suppression, changing to "undetectable viral load cannot pass HIV on through sex".

2. There is previous SR and MA has already been performed (eg. Mekuriaw, B., Belayneh, Z., Molla, A. et al. A meta-analysis and systematic review20, 55 (in 2021).https://doi.org/10.1186/s12954-021-00503-6). What is the difference between this previous work (alcohol use and its determinants among adults living with HIV/AIDS) and the current work (alcohol use on HIV risk behavior)? Please provide information about the heterogeneity of this previous MA.

3. More description of the "core problem" is required than justification.

4. "Moreover, there is no single country-level figure estimating the pooled prevalence of HIV risk behavior among PLWHA in Ethiopia." Take a look at # 2 for an example.

Methods:

1. Is there any amendment or changes? Did the authors write a protocol prior to doing this SR research (if any)? Additionally, there is a major difference in the description from the PROSPERO registration (CRD42020160018) e.g. The title at PROSPERO is "systematic review and meta-analysis on the magnitude and determinants of inconsistent condom use among adult HIV patients in Sub-Saharan Africa," which is very different from the title, which sets "HIV risk behavior and its association with alcohol intake, HIV status non-disclosure, and perceived stigma among people living with HIV/AIDS in Ethiopia: A systematic review and meta-analysis." Please provide an explanation and revision at PROSPERO.

2. Search strategies: Why did authors not use mesh terms in their search strategy? Moreover, the search strategies must be clearly stated in the annex (S2 File). You mention only PubMed, why not others? It should be clearly shown the search strategies by "PICO" or "PECO" or "CoCoP" and finally use Boolean terms like "AND" or "OR." Then put in the exact date and time of the search with its findings. How do you get gray literature?

3. Why did the authors collect studies only from 2000 to 2020? Shouldn't the result section have included data from 2008?

4. Authors should explain more data/information extraction selections in detail and place them in a table with the contents for the chosen article.

5. More information about the quality assessment of selected articles is required.

6. Authors concluded heterogeneity in their MA by using the I2. The I2 is a value that quantifies heterogeneity, but the presence of significant heterogeneity should be done with the Q test and its respective p-value.

7. The subgroup analysis, sensitivity analysis, and meta regression should be more explained in the data analysis method section as there is a finding based on this analysis in the result section.

8. Authors are encouraged to test for group differences and within the group subgroups as well (a p-value for interaction between groups).

Results

1. The "systematic review" was really rushed. More explanation and description of SR is required.

2. Please indicate the type of ES, e.g., figure 2.

Discussion

1. Please add more sentences on the implications of the main findings in this section.

2. Heterogeneity is very important across all analyses, even in the subgroup analyses. Authors are encouraged to discuss the manuscript's limit.

3. Is there any strength in the manuscript? Please explain this, e.g., adding a body of knowledge, a methodological perspective.

7. PLOS authors have the option to publish the peer review history of their article (what does this mean?). If published, this will include your full peer review and any attached files.

Reviewer #3: **Yes: **Ebisa Turi

Reviewer #4: **Yes: **Achenef Asmamaw Muche

---

## [Author Response · Author response to Decision Letter 1]

16 May 2022

Douglas S. Krakower, MD

RE: Manuscript ID: PONE-D-21-17808R1 (HIV risk behavior and its association with alcohol intake, HIV status non-disclosure, and perceived stigma among people living with HIV/AIDS in Ethiopia: A systematic review and meta-analysis.)

Dear Dr. Douglas S. Krakower,

Thank you very much for your email and the comments/suggestions of the reviewers and academic editor. We have looked at the comments and have revised our paper accordingly. We hope our paper improved as a result of incorporating the reviewers' and academic editor's comments and suggestions.

Please find for your kind consideration the following:

A rebuttal letter that responds to each point raised by the academic editor and reviewer is labeled 'Response to Reviewers'. The point-by-point responses are written in italic font style.

A revised manuscript with track changes labeled 'Revised Manuscript with Track Changes'.

A revised paper without tracked changes labeled 'Manuscript'

While hoping that these changes would meet with your favorable consideration, we are happy to hear if there are more comments and suggestions. Please do not hesitate to let us know if you have any questions.

Yours Sincerely,

Metadel Adane

Department of Environmental Health, Wollo University 

Dessie, Ethiopia

Tel:+251910336962

E-mail:metadel.adane2@gmail.com

Point by point response

Editor’s comment 

Comment 1: Addressing the most recent reviewers' comments that relate to the statistical methods used and their interpretation and discussion, to ensure methodological rigor in the final published manuscript.

Response: Thank you, dear editor. We have tried to address reviewers' comments accordingly. 

Comment 2: The term "HIV risk behavior" is currently used in an imprecise manner in the paper. Please be specific in the behaviors that you are describing. If you mean condom-less sex, sex with multiple partners, or some other behavior, please mention these in specific. "Unprotected" sex is not specific, as PLWH who have a suppressed viral load cannot transmit HIV to their sexual partners, so sex without condoms in this scenario is "protected" from HIV transmission (even if not from other STIs or pregnancy); please change to "condomless sex" if that is the intended meaning. 

Response: Thank you. Studies included in this meta-analysis did not address the same behavior. Some assessed risky sexual behavior, others assessed inconsistent condom use, condom unprotected sex, etc.….. So it is difficult to specify exactly those behaviors that are being addressed since the included studies assessed different behaviors. So the term that represents all these behaviors is risky sexual behavior. We used the term risky sexual behaviors to represent all these behaviors since risky sexual behavior according to the Center for Disease Control definition includes one or more of the following behaviors: sex without the use of condoms, inconsistent condom use, having multiple sexual partners, casual sex, sex with the influence of alcohol, and sexual exchange (exchange of money for sexual intercourse). Moreover, we have tried to use condomless sex to represent unprotected sex throughout the document.

Comment 3: In line 60, the term "premise" is not appropriate, as this suggests that the assertion that follows is uncertain or untested. Please change to: "There is robust evidence that PLWH who are undetectable on ART cannot transmit HIV to their sexual partners." 

Response: Thank you, dear reviewer. We have amended it accordingly. 

Comment 4: Lines 61 to 67 suggest that the acquisition of drug-resistant HIV strains among PLWH is a major factor in the need for second-line ART. The totality of the literature does not support this assertion. Further, the references cited are generally more than a decade old, which ignores a much more robust set of studies from 2010 onward that do not support the authors' statements. Without a more evidence-based and updated argument, the introduction is not appropriate for publication. It would be appropriate to state that "However, PLWH may have challenges in accessing and adhering to ART, which can result in detectable viral loads and the potential for HIV transmission" with the support of new references. 

Response: Thank you, dear reviewer. The comment is accepted and addressed accordingly. 

Comment 5: Lines 80-82: The authors do not justify why the factors listed are the "most fundamental to intervene," and this strong assertion needs to be moderated. It would be more appropriate to state that these factors merit consideration as ones that could impact sexual behaviors and might be amenable to interventions.

Response: Thank you for your constructive comment. We have amended it accordingly. 

Comment 6: Line 129: the authors need to justify why unpublished studies were included in the analyses.

Response: Thank you dear reviewer for remembering an important issue. All the studies included in this meta-analysis were published articles. However, during the initial searching process, we found one unpublished study which has been posted in the research square. Due to this reason, we have said that “both published and unpublished studies were included in the review”. But finally, when we checked its publication status, it was already published in the journal. So in this study, only published studies were included. We have tried to revise the inclusion criteria in our revised manuscript. 

Comment 7: Lines 136-139: The authors need to state clearly how they operationalized alcohol use given the heterogeneous definitions used in prior studies. Because the definition for this and other covariates of interest, such as HIV status disclosure, were heterogeneous, this needs to be discussed as a major limitation when considering the study findings in the Discussion. 

Response: Thank you, dear reviewer. We have tried to revise the operational definition and discuss it as a limitation in the discussion section accordingly. 

Comment 8: In the Discussion, as above, the authors need to be far more precise when discussing HIV risk behaviors, and must specify exactly those behaviors that are being addressed, as opposed to the general term of "HIV risk behaviors," which can mean many different things. Unless all of the studies in lines 278-285 addressed the same behaviors, then it is not appropriate to make quantitative comparisons across studies as the authors have done.

Response: Thank you for your constructive comment. As we have stated in the document, different literature defined risky sexual behavior as a behavior that includes one or more of the following behaviors: sex without the use of condoms, inconsistent condom use, having multiple sexual partners, casual sex, sex with the influence of alcohol, and sexual exchange (exchange of money for sexual intercourse). Studies included in this meta-analysis did not assess the same behaviors. Some assessed risky sexual behavior and others assessed inconsistent condom use or condom unprotected sex. So it is difficult to specify exactly those behaviors that are being addressed since they assessed different behaviors. Due to this reason, we used the term HIV risk behaviors to represent all these behaviors. We acknowledged it as a limitation in the revised manuscript. 

Comment 9: Line 39 and elsewhere: the term "safe sex" is outdated, non-specific, and not appropriate. Please remove this and specify exactly what is meant, such as how alcohol could influence decisions around condom use, status disclosure, or other factors that can affect HIV transmission. 

Response: Thank you, dear reviewer. We have amended it throughout the document. 

Comment 10: Line 347: The term "alcohol abuse" is stigmatizing; please change it to "alcohol use" or "alcohol use disorders" or similar. As above, please also remove "unsafe" as a term. 

Response: Thank you. The comment is accepted and addressed accordingly. 

Comment 11: The Discussion and Conclusion are missing a critical emphasis on the importance of promoting access and adherence to ART as a way to decrease HIV transmission, as suppressed viral loads are the most effective way to prevent transmission - even more than condoms or any social-behavioral interventions. The authors can discuss how counseling about alcohol, disclosure, and stigma is important to decrease HIV transmission among those without the use of ART, but ART needs to be mentioned front and center given the immense strength of evidence behind this biomedical strategy.

Response: Thank you. We have tried to revise the discussion and conclusion section accordingly. 

Reviewer #3

Comment 1: I read with great interest the Manuscript titled "Risky sexual practice and its association with alcohol intake, HIV status disclosure, and perceived stigma among adult HIV infected patients in Ethiopia. A systematic review and meta-analysis" which falls within the aim of PLOSE ONE Journal. In my honest opinion, the topic is interesting enough to attract the readers' attention. The methodology is accurate and conclusions are supported by the data analysis. Nevertheless, the authors should clarify some points and improve the operational definition of some variables.

Response: Thank you dear reviewer for your constructive comments. We have tried to revise the manuscript based on the comments of the reviewer accordingly.

Comment 2: The rationale that the authors stated is not consistent across the document. For instance, in the abstract, the authors said “ this meta-analysis aimed to estimate the pooled prevalence of the risky sexual practice and its association with……" but in the background section page 4, lines 80-81 the authors said, " However, we only examine the association between risky sexual behavior and alcohol intake, HIV status disclosure, and perceived stigma since they were the most frequently mentioned factors and have controversial findings among the included studies”.

Response: Thank you. We have amended it accordingly.

Comment 3: How the authors operationalized alcohol consumption, perceived stigma, and HIV status disclosure is not clear. Their definition across the studies might be different and the authors should add one table concerning the definition of these variables in each study included in this review.

Response: Thank you for your suggestion. The definition of perceived stigma and HIV status disclosure is similar across the included articles due for this reason, there is no need to put their definition in table 1. However, we have tried to put the definition of alcohol use across the studies in table 1 of our revised manuscript. 

Comment 4:In the abstract section, the authors said “In Ethiopia, findings regarding risky sexual practice have been inconsistent and inconclusive". However, they said nothing about this in the background section. Please, show us the inconsistency you found by elaborating on this section.

Response: Thank you. We have tried to show the inconsistencies in the background section accordingly.

Reviewer #4: 

Comment 1: This is a good manuscript, clearly written, with the correct methodology and correct statistics for meta-analysis. However, it needs revision to be improved.

Response: Thank you, dear reviewer. We have tried to revise our manuscript according to your comment.

Comment 2: Tile: It was too long! "It’s association with alcohol intake, HIV status non-disclosure, and perceived stigma" This listed associated factor can be written concisely.

Response: Thank you. We have modified the title to "HIV risk behavior and associated factors among people living with HIV/AIDS in Ethiopia: A systematic review and meta-analysis.”

Abstract:

Comment 3:International databases such as Google Scholar, the Cochrane Library, HINARI, PubMed, CINAHL, and Global Health. Good to identify which one is a database and which one is a search engine? Furthermore, why are you using HINARI and Global Health? It is good to consider if you consider others, i.e., AJOL, WorldCat

Response: Thank you, dear reviewer. Except for google scholar; Cochrane Library, HINARI, PubMed, CINAHL, and Global Health are databases. We used HINARI and Global Health since they are easily accessible (Wollo university has the password for the HINARI database due to this we can easily access other databases via HINARI). We have already missed AJOL and WorldCat in the current meta-analysis. We will consider them for our future experience. 

Comment 4:Why did you use the word "prevalence" for "HIV risk behavior"?

Response: Thank you, dear reviewer. Some studies included in this study used the word "prevalence". So, we used this term in order not to miss those studies. 

Background

Comment 5: "There is also a premise that PLWHA who achieve and maintain viral load suppression cannot transmit HIV to their HIV-negative sexual partners [11]." I suggested instead of maintaining viral load suppression, changing to "undetectable viral load cannot pass HIV on through sex".

Response: Thank you for your suggestion. We have amended it accordingly. 

Comment 6:There is previous SR and MA has already been performed (eg. Mekuriaw, B., Belayneh, Z., Molla, A. et al. A meta-analysis and systematic review20, 55 (in 2021).https://doi.org/10.1186/s12954-021-00503-6). What is the difference between this previous work (alcohol use and its determinants among adults living with HIV/AIDS) and the current work (alcohol use on HIV risk behavior)? Please provide information about the heterogeneity of this previous MA.

Response: Thank you. There is a big difference between the previous work and the current one. The outcome variable for the previous work was alcohol use among PLWHA and the outcome variable for the current study is HIV risk behavior. Alcohol use was used as an independent predictor of HIV risk behavior in the current study as opposed to the previous work which was used as an outcome variable. 

Comment 7: More description of the "core problem" is required than justification.

Response: Thank you. The comment is accepted and addressed accordingly. 

Comment 8: "Moreover, there is no single country-level figure estimating the pooled prevalence of HIV risk behavior among PLWHA in Ethiopia." Take a look at # 2 for an example.

Response: Thank you for your suggestion. We have amended it accordingly.

Methods:

Comment 9:Is there any amendments or changes? Did the authors write a protocol before doing this SR research (if any)? Additionally, there is a major difference in the description from the PROSPERO registration (CRD42020160018) e.g. The title at PROSPERO is "systematic review and meta-analysis on the magnitude and determinants of inconsistent condom use among adult HIV patients in Sub-Saharan Africa," which is very different from the title, which sets "HIV risk behavior and its association with alcohol intake, HIV status non-disclosure, and perceived stigma among people living with HIV/AIDS in Ethiopia: A systematic review and meta-analysis." Please provide an explanation and revision at PROSPERO.

Response: Thank you for your constructive comment. Initially, we proposed to do SR research entitled "Systematic review and meta-analysis on the magnitude and determinants of inconsistent condom use among adult HIV patients in Sub-Saharan Africa," which was registered at PROSPERO with a registration number of CRD42020160018. However, due to the small number of articles, we have changed the title to "HIV risk behavior and its association with alcohol intake, HIV status non-disclosure, and perceived stigma among people living with HIV/AIDS in Ethiopia: A systematic review and meta-analysis.” We used the registration number of the former title to the current one without amending the protocol which may mislead the reader. We acknowledged the problem and have removed the registration status of the title from our manuscript. 

Comment 10: Search strategies: Why did the authors not use mesh terms in their search strategy? Moreover, the search strategies must be clearly stated in the annex (S2 File). You mention only PubMed, why not others? It should be clearly shown the search strategies by "PICO" or "PECO" or "CoCoP" and finally, use Boolean terms like "AND" or "OR." Then put in the exact date and time of the search with its findings. How do you get gray literature?

Response: Thank you for your comment. We have amended the search strategy and tried to put the findings from each searching database in S2 File accordingly. Gray literature were identified by traditional Google search and Addis Ababa digital library search. 

Comment 11: Why did the authors collect studies only from 2000 to 2020? Shouldn't the result section have included data from 2008?

Response: Thank you. We have tried to collect articles published from the beginning of 2000 to September 30, 2020, to get more articles assessing HIV risk behavior. However, during the searching processes, we found only articles published from 2008-to 2020. 

Comment 12:Authors should explain more data/information extraction selections in detail and place them in a table with the contents of the chosen article.

Response: Thank you, dear reviewer. We have tried to include more data in the data extraction section accordingly. However, we haven’t seen the advantage of putting the data extraction section in tablet form since it is already attached as supporting information. 

Comment 13:More information about the quality assessment of selected articles is required.

Response: Thank you. We have modified the quality assessment section according to your comment.

Comment 14:Authors concluded heterogeneity in their MA by using the I2. The I2 is a value that quantifies heterogeneity, but the presence of significant heterogeneity should be done with the Q test and its respective p-value.

Response: Thank you for your constructive comment. Although Q test is the usual way of assessing whether a set of single studies are homogeneous in the meta-analysis, it only informs us of the presence versus the absence of heterogeneity, but it does not report on the extent of such heterogeneity. In addition, Q has low power as a comprehensive test of heterogeneity especially when the number of studies is small (Gavaghan et al, 2000). Recently, the I2 index has been proposed to quantify the degree of heterogeneity in a meta-analysis. A study which is highlighted in yellow color (Huedo-Medina TB, Sánchez-Meca J, Marin-Martinez F, Botella J. Assessing heterogeneity in meta-analysis: Q statistic or I² index?. Psychological methods. 2006 Jun; 11(2):193) showed that the I2 index as a complement to the Q test due to this reason we can use I2 and its respective p-value to indicate the presence of significant heterogeneity

Comment 15:The subgroup analysis, sensitivity analysis, and Meta-regression should be more explained in the data analysis method section as there is a finding based on this analysis in the result section.

Response: Thank you, dear reviewer. We have explained them in the data analysis section accordingly. 

Comment 16:Authors are encouraged to test for group differences and within the group subgroups as well (a p-value for interaction between groups).

Response: Thank you. Have tried to test for group differences and within the group subgroups among the included studies. In addition, we have to put a p-value for interaction between groups in table 2 of our revised manuscript. 

Results

Comment 17:The "systematic review" was really rushed. More explanation and description of SR is required.

Response: Thank you. The comment is not clear for as. As much as possible, we have tried to put the detail explanation and description of each content of the systematic review in our manuscript. 

Comment 18: Please indicate the type of ES, e.g., figure 2.

Response: Thank you. The comment is accepted and addressed accordingly in fig 2. 

Discussion

Comment 19: Please add more sentences on the implications of the main findings in this section.

Response: Thank you. We have tried our best to add the implications of the main findings in the discussion section accordingly. 

Comment 20: Heterogeneity is very important across all analyses, even in the subgroup analyses. Authors are encouraged to discuss the manuscript's limit.

Response: Thank you. We have tried to discuss it as a limitation in the discussion section. 

Comment 21:Is there any strength in the manuscript? Please explain this, e.g., adding a body of knowledge, and a methodological perspective.

Response: Thank you. We have tried to put the strength of our study in the discussion section.

---

## [Editor Report · Decision Letter 2]

19 May 2022

HIV risk behavior and associated factors among people living with HIV/AIDS in Ethiopia : A systematic review and meta-analysis.

PONE-D-21-17808R2

Dear Dr. Adane,

We’re pleased to inform you that your manuscript has been judged scientifically suitable for publication and will be formally accepted for publication once it meets all outstanding technical requirements.  Please also address my comments to the authors before finalizing the manuscript.

Kind regards,

Douglas S. Krakower, MD

Academic Editor

PLOS ONE

Additional Editor Comments (optional):

1. Please proofread carefully for typos, which occur in the abstract and manuscript.

2. Line 53: please specify that you are referring to the US Centers for Disease Control and Prevention.

3. In Lines 83 to 88, please specify what the percentages represent; is it the percent of PLWH who have engaged in an HIV risk behaviors ever, or during some particular time frame?
---

## [Editor Report · Acceptance letter]

29 Jun 2022

PONE-D-21-17808R2 

HIV risk behavior and associated factors among people living with HIV/AIDS in Ethiopia: A systematic review and meta-analysis 

Dear Dr. Adane:

I'm pleased to inform you that your manuscript has been deemed suitable for publication in PLOS ONE. Congratulations! Your manuscript is now with our production department. 

Kind regards, 

on behalf of

Dr. Douglas S. Krakower 

Academic Editor

PLOS ONE